# Seroprevalence of hepatitis E virus infection in the Americas: Estimates from a systematic review and meta-analysis

**Nathalie Verónica Fernández Villalobos**[1]*, **Barbora Kessel**[2], **Isti Rodiah**[2], **Jördis Jennifer Ott**[2,3], **Berit Lange**[2,4], **Gérard Krause**[2,3,4,5]

1 Department of Epidemiology, PhD Programme, Helmholtz Centre for Infection Research (HZI), Braunschweig-Hannover, Germany, 2 Department of Epidemiology, Helmholtz Centre for Infection Research (HZI), Braunschweig, Germany, 3 Hannover Medical School, Hannover, Germany, 4 German Centre for Infection Research (DZIF), Braunschweig, Germany, 5 Twincore, Centre for Experimental and Clinical Infection Research, Braunschweig-Hannover, Germany

* Nathalie.Fernandez@helmholtz-hzi.de

## Abstract

### Background

Hepatitis E virus (HEV) infection is responsible for inflammatory liver disease and can cause severe health problems. Because the seroprevalence of HEV varies within different population groups and between regions of the continent, we conducted a systematic review on the topic in order to provide evidence for targeted prevention strategies.

### Methods

We performed a systematic review in PubMed, SCIELO, LILACS, EBSCO, and Cochrane Library and included reports up to 25 May 2021 (PROSPERO registration number: CRD42020173934). We assessed the risk of bias, publication bias, and heterogeneity between studies and conducted a random-effect meta-analysis for proportions using a (binomial-normal) generalized linear mixed model (GLMM) fitted by Maximum Likelihood (ML). We also reported other characteristics like genotype and risk factors.

### Results

Of 1212 identified records, 142 fulfilled the inclusion criteria and were included in the qualitative analysis and 132 in the quantitative analysis. Our random-effects GLMM pooled overall estimate for past infection (IgG) was 7.7% (95% CI 6.4%–9.2%) with high heterogeneity ($I^2$ = 97%). We found higher seroprevalence in certain population groups, for example in people with pig related exposure for IgG (ranges from 6.2%–28% and pooled estimate of 13.8%, 95% CI: 7.6%–23.6%), or with diagnosed or suspected acute viral hepatitis for IgM (ranges from 0.3%–23.9% and pooled estimate of 5.5%, 95% CI: 2.0%–14.1%). Increasing age, contact with pigs and meat products, and low socioeconomic conditions are the main risk factors for HEV infection. Genotype 1 and 3 were documented across the region.

**Data Availability Statement:** All relevant data are within the paper and its Supporting information files.

**Funding:** The funders had no role in study design, data collection and analysis, decision to publish, or preparation of the manuscript.

**Competing interests:** The authors have declared that no competing interests exist.

## Conclusion

HEV seroprevalence estimates demonstrated high variability within the Americas. There are population groups with higher seroprevalence and reported risk factors for HEV infection that need to be prioritized for further research. Due to human transmission and zoonotic infections in the region, preventive strategies should include water sanitation, occupational health, and food safety.

## Introduction

Hepatitis E virus (HEV) is responsible for a liver disease that affects millions of people worldwide, especially in low- and middle-income countries [1]. In 2005, a disease burden of 20 million HEV infections worldwide and 3.3 million symptomatic cases was estimated [2]. For 2019, the Institute for Health Metrics and Evaluation has also calculated a global HEV prevalence rate of 20.5 (95% uncertainty intervals 16.8–24.7) cases per 100 000 people [3].

Generally, the HEV disease is self-limiting and has mild symptoms; however, in some cases it can result in severe acute hepatitis, extrahepatic disorders, chronic hepatitis leading to cirrhosis, and fulminant hepatitis [4]. Pregnant women have been documented to have an increased risk of fulminant hepatitis [5]. Chronic hepatitis E infection has been reported in immunocompromised people [6], affecting mainly solid organ transplants (SOT) patients with 66% development of chronic hepatitis [7].

HEV genome can be grouped into 8 genotypes (HEV-1 to HEV-8), but only genotype 1–4 can infect humans [8]. Genotype 1 and 2 have been identified mainly in Asia and Africa, and they are related to fecal-oral transmission; while, genotype 3 and 4 are identified as zoonosis and circulate in animals and humans. Genotype 3 has been documented worldwide [8].

Current literature suggests that the main route of transmission of the virus is fecal-oral, mainly through contaminated drinking water or food, and it is commonly found in low- and middle-income countries due to the limited access to drinkable water and poor sanitation infrastructure [6].

Most systematic reviews on HEV focus on Europe [9], the Middle East, and the North African region [10]. Some reviews for HEV seroprevalence included individual countries of the Americas [11, 12], but not the whole continent. A review at global level also included some countries of the Americas [13], but it did not include all available evidence on the situation on the continent. Other reviews have focused on non-endemic countries [14, 15], industrialized countries [16], or on a specific population such as children [17], immunocompromised individuals [18], or pregnant women [19] (Table 2, page 3 in S1 File). Thus, there is a lack of systematically retrieved evidence on the seroprevalence and risk factors of HEV in the majority of the countries in the Americas, and there is not a realistic picture of the HEV seroprevalence in the continent. There is a need to compile evidence that includes different populations, diagnostic methods, and especially considering particular languages and databases of the region.

In this systematic review, we aimed to estimate the HEV seroprevalence of the Americas focusing on the analysis of serological markers, population groups, and regions. We also considered the risk factors, genotypes, and routes of transmission associated with the infection of HEV in the continent. Likewise, the aim was to provide evidence for improving the related burden of this disease and to target prevention strategies for HEV.

## Material and methods

### Search strategy

We performed a systematic review (registration number in PROSPERO CRD42020173934), following the PRISMA guidelines and checklist (S2 File) [20], in PubMed, SCIELO, LILACS, EBSCO, and Cochrane Library, searching for publications on seroprevalence of Hepatitis E Virus (search terms "hepatitis E" and "seroprevalence," page 1 in S1 File) without date or language restrictions. The literature search included reports up to 25 May 2021. Further, we applied the snowball method [21] starting with the references of the existing reviews to ensure that we included all available relevant studies. All references were processed in Mendeley Desktop [22].

### Inclusion and exclusion criteria

We included reports conducted in the Americas, reporting anti-HEV immunoglobulin G, M, and/or total (IgG, IgM, and/or total anti-HEV), and Ribonucleic acid (RNA) in humans. We defined Americas as the countries included in the Pan American Health Organization (PAHO), and we added one study that was done in Greenland in Inuits; a population that inhabits the Arctic regions of Greenland, Canada, and Alaska [23].

We considered the different serological markers based on the detection of anti-HEV antibodies: i) IgM markers of acute or recent infection, ii) IgG markers of past infection, iii) total antibodies markers of detection of antibodies to HEV, and iv) RNA markers of acute infection.

We included publications using any of the following study designs: observational studies, surveys, cohort studies, cross-sectional studies, case control studies, and outbreak investigation reports. We excluded studies reporting seroprevalence based on fecal samples.

### Data extraction

One researcher (NVFV) extracted the following information: first author, journal, year of publication, country, sample size for total anti-HEV, sample size for IgG, sample size for IgM, sample size for RNA, start and end of data collection, assay used for total anti-HEV, assay used for IgG, assay used for IgM, assay used for RNA, seroprevalence values total anti-HEV, seroprevalence values IgG, seroprevalence values IgM, seroprevalence values RNA, unit of measure of effect (outcome = odds ratio (OR) and relative risk (RR)), genotype, population group, age, the proportion of male, risk factors (outcome = list of findings), routes of transmission (outcome = list of findings). Two other researchers (BK & IR) independently crosschecked all extracted data. Any conflicts were resolved in a discussion.

### Risk of bias assessment

Two researchers (NVFV & IR) assessed the risk of bias using an adapted version of the ROBINS-I tool [24] for non-randomized studies. Specifically, they analyzed the studies in terms of bias due to confounding (high risk if no age information was provided), selection of participants and follow-up (high risk if different follow-up time for different groups was performed), misclassification of exposure (high if the source was unclear), missing data (high if participants were excluded on large scale for missing data), measurement of outcome (high if the report was not available), or reporting (high if selective subgroup was reported). We measured the risk scales as low, moderate, and high. Any conflicts in the assessment were resolved in a discussion.

## Data analysis

**Descriptive.**  We performed a qualitative synthesis with all the included studies and presented their main characteristics as tables and as narrative. Moreover, we organized the findings according to the population groups and regions of the Americas (North America, South America, Central America and the Carribean). Further, we grouped different population groups as follows: 1) general population, 2) blood donors, 3) ethnic groups, 4) exposed population, 5) occupational group, 6) pig related occupation, 7) viral hepatitis, 8) immunodeficiency, 9) pregnant women, 10) rural population, 11) children, and 12) combined.

For those groups that were not consistent, we applied the following definitions:

- General population: urban, mixed urban, and rural residents, civilian participants, people without exposure, volunteers, and controls.

- Ethnic groups: Indigenous population, Indians population, tribes, and Amazonian and Andean population.

- Exposed population: Cohabitants of the exposed people, inmates, homeless people, alcoholics, sex workers, persons who inject drugs, homosexuals, and travelers.

- Occupational group: veterinary students and workers, healthcare workers, waste pickers, religious groups, workers of natural resources, biologists, hunters, workers in water and sewage companies, soldiers, and people deployed in the military.

- Pig related occupation: Swine related occupations and people who work in pig farms with direct contact with pigs.

- Viral hepatitis: People with acute hepatitis, acute jaundice, positive anti-hepatitis antibodies, suspected cases, sporadic cases, and outbreak cases.

- Immunodeficiency: Hemodialysis patients, people living with human immunodeficiency virus (HIV), transplant or plasma recipients, patients with chronic hepatitis C virus (HCV) or hepatitis B virus (HBV), people with liver failure, cancer, Elevated alanine aminotransferase (ALT), patients with plasmapheresis, patients with hepatosplenic schistosomiasis, patients with *Schistosoma mansoni*, and people subjected to invasive procedures.

From the included studies, we also extracted information related to HEV genotypes found in the region and geographically represented their distribution in the Americas using vector layers based on shapes from Environmental Systems Research Institute (ESRI) [25] and using R [26] version 4.0.2 (using packages "ggplot2" [27], "sf" [28], "dplyr" [29]).

**Publication bias.**  We used funnel plots from the R package "meta" [30] to visualize the interplay between reported seroprevalence estimates and their precision in order to identify any irregularities, which we then interpreted according to the recommendations by Sterne et al. [31]. We considered the risk of not publishing a result because a particular seroprevalence value has been found to be low in our setting. However, it potentially had a considerable impact on our results due to the number of studies available per country and per population group. Therefore, we visualized the composition of the studies by stacked bar charts (constructed in "ggplot2" [27] package in R) and accounted for it when interpreting the results.

**Meta-analysis.**  We conducted our analysis for each of the considered markers, total anti-HEV, IgG, IgM, RNA, separately. IgM and RNA results were further separated into those obtained from the whole sample and those obtained from previously positive samples (for example IgG or total anti-HEV positive). For RNA in positive samples, we refrained from conducting a meta-analysis due to low sample sizes (only two studies with more than 50 samples)

and varying definition of "previously positive samples" (IgG positive, IgM positive, at least one of the before mentioned, or both at the same time).

In all other cases, using the raw data (the reported number of positive and the reported sample size) for each study, we performed a random-effect meta-analysis for proportions using a (binomial-normal) generalized linear mixed model (GLMM) fitted by Maximum Likelihood (ML). We constructed a 95% confidence interval for the pooled estimate using t-quantiles with k-1 degrees of freedom (k = number of pooled studies). We assessed heterogeneity visually by forest plots, by assessing the percentage of variance over the studies using $I^2$, and by estimating the between study variance $tau^2$.

We performed subgroup analyses by region, country, and population group. Based on previous knowledge and the observed large heterogeneity between the reported seroprevalences, we expected different seroprevalence levels in some population subgroups. For a formal comparison of two, or more pooled estimates, we used the Q-test for subgroup differences as implemented in the R package "meta" [30] (referred to also as "comparison based on fixed-effects model between subgroups" [32] in literature).

In three instances the reported seroprevalence included in the pooling was based on the National Health and Nutrition Examination Survey (NHANES) dataset, and thus subject to necessary corrections due to the survey design. To be able to include these seroprevalences into the random-effects GLMM framework, we calculated for each of them a fictive number of positive cases and a fictive sample size that would yield exactly the reported seroprevalence, and the reported 95% confidence interval assumed to be constructed on the logit-scale using normal quantiles. The resulting decimal numbers were then rounded when calling the *meta-prop* function in package "meta" [30].

In each meta-analysis, we included one seroprevalence result per sample. Our choice is detailed in Table 4, pages 9–36 in S1 File; in Fig 3, page 62 in S1 File (for samples analyzed repeatedly by different tests); and in Fig 6, page 65 in S1 File (for studies based on the NHANES dataset).

In case of samples repeatedly analyzed by different tests, we examined the impact of our choice by conducting a sensitivity analysis. We replaced the primarily chosen values by their alternatives one at a time and all at once, and carried out the respective meta-analysis. When pooling within regions involved several seroprevalences reported in one publication, we examined whether allowing for a positive correlation between such results would change our pooled estimates. We did this by endowing the basic random-effects GLMM with an additional random effect (for publication) and fitted the extended model using R package "lme4" [33].

All the analysis were performed using R [26] version 4.0.2 (using package "meta"[30] and "metaphor" [34]). Forest plots were created using R package "forestplot" [35]. The significance level used was 0.05.

## Results

We identified 1212 records and retrieved 217 for full-text screening. After verifying the inclusion criteria, we included 142 for our qualitative analysis and 132 for the quantitative analysis (Fig 1). The snowball searching of references in the existing reviews did not reveal any additional relevant publications.

Of the 142 included articles, the majority of studies originated from Brazil (n = 38), United States (USA; n = 33), and Argentina (n = 14). According to region, South America hosted 57% of the included studies (n = 81), North America 37% (n = 52), and Central America and the Caribbean 6% (n = 9). The studies were conducted between 1967 (retrospective) and 2018,

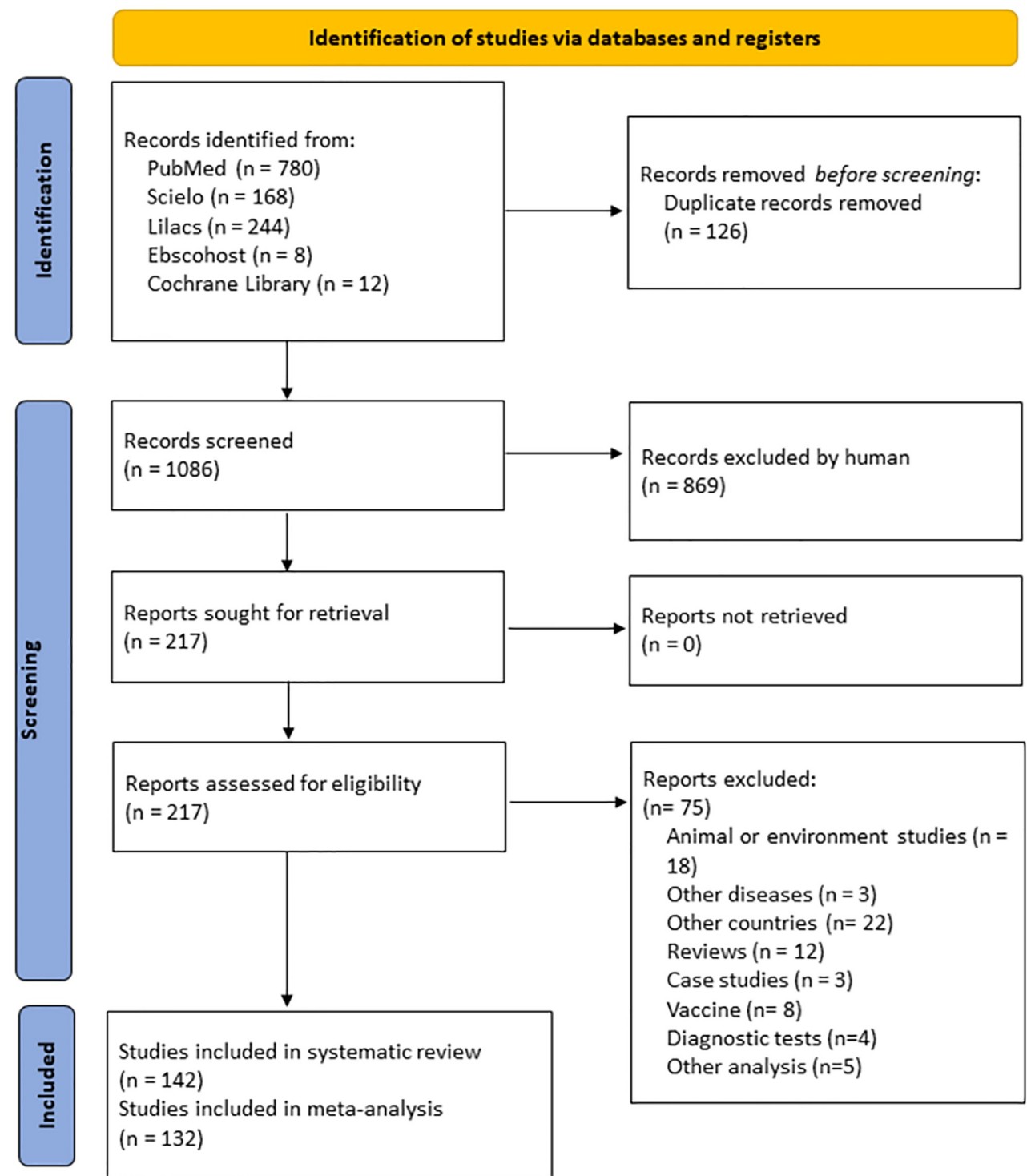

**Fig 1. Flow diagram of the selected studies for the HEV seroprevalence.**

and they included different population groups. The sample sizes ranged between 14 and 18 829 participants, with ages between 1–98 years.

Regarding the assays, 20% of the studies performed the analysis with Abbott Laboratories (n = 29), followed by the combination of multiple assays with 15% (n = 21), and by "in house" ELISA (enzyme-linked immunosorbent assay) methods with 14% (n = 20). The characteristics of the included studies are presented in the Table 4, page 9–36 in S1 File.

## Risk of bias assessment

We assessed the risk of bias due to a) confounding, b) selection, c) misclassification, d) missing data, and e) measurement of outcome. Confounding was moderate in most studies as adjusted estimates or age information was provided. High risk of bias was found for 18 studies as they did not report any age related information. Selection bias and missing data were mostly moderate, and misclassification was mostly low. The total assessment can be found in the Table 3, page 4–8 in S1 File. Since the majority of studies had a low or moderate risk assessment, we did not exclude any study and included all of them into the qualitative analysis.

## HEV seroprevalence

The results of our pooled analyses and the $I^2$ results by population group and by region are presented in the Tables 1 and 2, respectively. Further details can be found in supplementary figures as detailed in the text below in S1 File.

**Total HEV antibodies.** In 21 studies that reported on total anti-HEV antibodies (37 results), we found ranges of seroprevalence from 0%–40.6%. The random effects meta-analysis pooled estimate was 4.3% (95% CI 2.9%–6.4%) with high heterogeneity ($I^2$ = 95%).

The subgroup analysis (Table 1) shows that seroprevalences reported for some of the considered subpopulations tend to be higher. For example, exposed populations [seroprevalence range: 5.9%– 13.5% (n = 3), and pooled estimate: 9.6% (95% CI: 3.9%–21.9%)], people with pig related exposure [range: 10.9%– 40.6% (n = 3), and pooled estimate: 19.7 (95% CI: 3.8%– 60.2%)], or ethnic groups [range: 5.4%– 17% (n = 4), and pooled estimate: 8.5% (95% CI: 4.4%– 16%)] had higher seroprevalence compared with those reported for pregnant women

**Table 1. Pooled analysis of HEV seroprevalence by population group.**

| Population Group | Total anti-HEV | | | IgG | | | IgM (total sample) | | | IgM (+ sample) | | |
|---|---|---|---|---|---|---|---|---|---|---|---|---|
| | Proportion (95% CI) | $I^2$ | k* | Proportion (95% CI) | $I^2$ | k* | Proportion (95% CI) | $I^2$ | k* | Proportion (95% CI) [†] | $I^2$ | k* |
| General population | 3.1% (1.4%-6.8%) | 89% | 5 | 7.2% (5.2%-9.9%) | 93% | 21 | - | - | - | - | - | - |
| Blood donors | 4.2% (1.6%-10.9%) | 98% | 7 | 6.6% (4.2%-10.3%) | 98% | 28 | 0.8% (0.3%-2%) | 84% | 5 | 15.5% (4.2%-43.5%) | 79% | 6 |
| Ethnic groups | 8.5% (4.4%-16%) | 83% | 4 | 5.7% (2.7%-11.7%) | 94% | 7 | - | - | - | - | - | - |
| Exposed population | 9.6% (3.9%-21.9%) | 60% | 3 | 11.4% (5.4%-22.6%) | 96% | 12 | - | - | - | - | - | - |
| Occupational group | 5.3% (2%-13.7%) | 92% | 7 | 7.1% (3.7%-13.2%) | 68% | 11 | - | - | - | - | - | - |
| Pig related Occupation | 19.7% (3.8%-60.2%) | 93% | 3 | 13.8% (7.6%-23.6%) | 92% | 6 | - | - | - | - | - | - |
| Known viral hepatitis | - | - | - | 8.2% (4.6%-14.3%) | 97% | 21 | 5.5% (2.0%-14.1%)[‡] | 93% | 11 | 29% (16.2%–46%) | 32% | 4 |
| Immunodeficiency | - | - | - | 10.4% (6.8%-15.4%) | 96% | 30 | 0.9% (0.3%-2.4%) | 59% | 10 | | | |
| Pregnant women | 0.8% (0.2%-3.1%) | 19% | 4 | 3.0% (0.7%-11.4%) | 93% | 7 | - | - | - | - | - | - |
| Rural population | - | - | - | 11.5% (5.4%-22.9%) | 97% | 10 | - | - | - | - | - | - |
| Children | - | - | - | 1.1% (0.1%-8.1%) | 88% | 4 | - | - | - | - | - | - |

* Number of observations. Importantly, some studies included more than one population group, so this number is different than number of studies.

[†] Only studies with sample size at least 10 are meta-analyzed.

[‡] Without an outbreak with seroprevalence 100%.

**Table 2. Pooled analysis of HEV seroprevalence by region.**

| Region | Total anti-HEV | | | IgG | | | IgM (total sample) | | | IgM (+ sample) | | |
|---|---|---|---|---|---|---|---|---|---|---|---|---|
| | Proportion (95% CI) | $I^2$ | $k^*$ | Proportion (95% CI) | $I^2$ | $k^*$ | Proportion (95% CI) | $I^2$ | K | Proportion (95% CI)† | $I^2$ | $k^*$ |
| North America | 3.4% (1.5%-7.2%) | 98% | 11 | 9% (6.6%-12.2%) | 97% | 52 | 0.8% (0.4%-1.5%) | 77% | 13 | 2.5% (0.1%-48.1%) | 0% | 7 |
| South America | 4.1% (2.5%-6.5%) | 87% | 21 | 7.2% (5.7%-9.1%) | 96% | 97 | 1.5% (0.6%-3.8%) | 79% | 9 | 13.8% (7.7%-23.6%) | 67% | 17 |
| Central America and the Caribbean | 10.7% (2.4%-36.4%) | 93% | 5 | 6.5% (1.9%-20.3%) | 96% | 8 | - | - | - | - | - | - |

\* Number of observations. Importantly, some studies included more than one population group, so this number is different than number of studies.

† Only studies with a sample size of at least 10 were meta-analyzed.

[range: 0%–1.6% (n = 4), and pooled estimate: 0.8% (95% CI: 0.2%–3.1%)]. In many of the subgroups, the heterogeneity of the reported results remained high. See also Fig 4, page 63 in S1 File.

Regarding regions, 11 studies were from South America, nine studies were from North America (only one study outside of USA), and two in Central America and the Caribbean, with seroprevalence ranges from 0%–17%, 0.4%–13.5%, and 2.9%–40.6, respectively. All three regions show similar level of variability in the reported seroprevalences; a formal test does not confirm any significant differences between the pooled estimates (Q-test, p = 0.1678). For more details, see Table 2 and Fig 5, page 64 in S1 File.

**Past hepatitis E exposure (IgG).** Disregarding repeated analyses of the same sample, as described in Methods, we considered 157 seroprevalences reported in 105 publications. The seroprevalence ranges from 0%–71% (median of 8%), and the random-effects GLMM pooled overall estimate was 7.7% (6.4%–9.2%) with high heterogeneity ($I^2$ = 97%, tau$^2$ = 1.4).

Fig 2 shows an overview of the pooled results within subgroups, and details of the seroprevalences underlying the pooled estimates are given in the S1 File, pages 66–71.

Compared to general (adult) population (pooled estimate 7.2%, 95% CI: 5.2%–9.9%), children had a lower seroprevalence (1.1%, 95% CI: 0.1%– 8.1%, Q test p = 0.0037), and people with a pig related exposure had a higher one (13.8%, 95% CI: 7.6%– 23.6%, Q -test p = 0.0183). The heterogeneity in the subgroups remained high.

By region, 63 studies (97 results) were from South America, 36 (52 results) from North America, and six (eight results) from Central America and the Caribbean. Brazil (33 studies), the USA (20 studies), Argentina (13 studies), and Mexico (10 studies) were the countries with more number of published studies.

The seroprevalence by region ranged from 0%–66.3%, 0%–43.6%, and 0%–71% in South America, North America, and Central America and the Caribbean respectively. The pooled estimates (see Fig 2) were not statistically significantly different (Q-test, p = 0.4797). When we compared the regions within subgroups with more evidence (for example in blood donors, general population, viral hepatitis, and immunodeficiency), the collected data suggested a regional difference (between North and South America) only in the latter case. Details can be found in S1 File, pages 66–69.

Fig 2 shows also the pooled seroprevalences by country. They were mostly comparable with perhaps the exception of Peru, Bolivia, and Colombia (Fig 14, page 73 in S1 File). The heterogeneity within countries remained high (Fig 13, page 72 in S1 File).

**Acute hepatitis E (IgM).** Due to some studies that reported IgM results based on positive samples, we divided this group into two: those that use a total sample and those that used only samples with positive results.

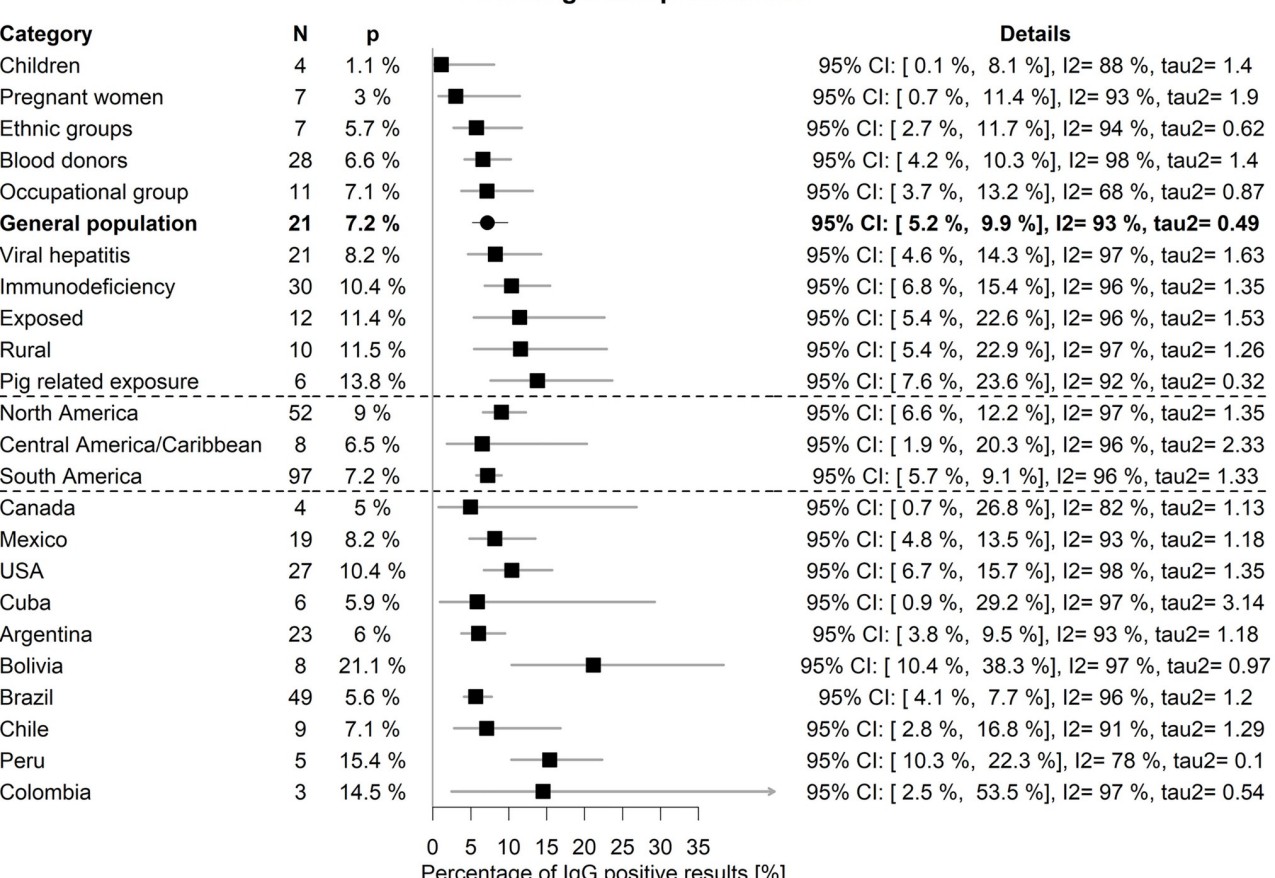

**Fig 2. Overview of pooled IgG seroprevalences obtained from random-effects GLMM with 95% confidence intervals constructed using t-quantiles.** N denotes the number of published results underlying the pooling. Further details behind the pooled numbers can be seen in pages 66–73 in S1 File. The very large heterogeneity among the results from Cuba is due to one study reporting a large seroprevalence of 71% for a sample of outbreak and sporadic acute cases of viral hepatitis. Without this result, the pooled estimates would change as follows: 3.0%, 95% CI: 1.0%–8.4%, $I^2$ = 94%, tau2 = 0.66 (Cuba) and 4.2%, 95% CI: 1.9%– 9.0%, $I^2$ = 93%, $tau^2$ = 0.70 (Central America/Caribbean).

*The total sample for IgM analysis.* Without including the viral hepatitis group, 23 IgM seroprevalences were reported by 21 studies (Fig 15, page 74 in S1 File). The observed seroprevalences ranged from 0% to 10% (median 1%). Values above 5% were reported for pediatric patients in Mexico [36] (6.1%, n = 99), ethnic groups in Argentina [37] (8.7%, n = 126), and female blood donors in Brazil [38] (10%, n = 20). A pooled seroprevalence was 0.9% (95% CI: 0.6%–1.5%, $I^2$ = 83%, $tau^2$ = 0.97).

In 12 studies reporting IgM results from sera of patients with confirmed or suspected acute viral hepatitis (viral hepatitis group), the seroprevalence ranged from 0% to 100%, with a median of 8%. Seroprevalences higher than 10% were reported from Venezuela [39], Chile [40], Colombia [41], and Cuba [42–44], the latter country accounted also for the observed prevalence of 100% [44] (Fig 16, page 75 in S1 File). Without this apparent HEV outbreak, the pooled seroprevalence was 5.5% (95% CI: 2.0%– 14.1%, $I^2$ = 93%, $tau^2$ = 2.1), significantly (Q-test, p = 0.006) higher than in the other subgroups (pooled together).

Comparing South and North America, we did not find a significant difference between the pooled estimates (Q-test, p = 0.2211), the former being 1.5% (95% CI: 0.6%–3.8%), the latter

0.8% (95%CI: 0.4%–1.5%) (Fig 17, page 75 in S1 File). Central America and the Caribbean was not considered in the comparison due to the low number of reported results (n = 1).

*IgM among positive samples*. In 26 studies that presented 33 results regarding IgM among IgG or total anti HEV positive cases, we found ranges of seroprevalence from 0%–100%, with median 16%. Considering only results based on a sample size of at least 10, the range was 0%–50%.

The pooled seroprevalence was 11.4% (95% CI: 6.2%–19.9%, $I^2$ = 55%, $tau^2$ = 1.8), split into subgroups we got 29% (95% CI: 16.2%–46%, $I^2$ = 79%, $tau^2$ = 1.5) for the viral hepatitis group, and 15.5% for blood donors (95% CI: 4.2%–43.5%, $I^2$ = 32%, $tau^2$ = 0) (Fig 18, page 76 in S1 File).

By region, 19 studies were performed in South America (Brazil = seven, and Argentina = seven), six in North America (USA = five), and one in Central America and the Caribbean (Fig 19, page 77 in S1 File). The data did not confirm differences between South and North America (p = 0.218 for the subgroup comparison), with a pooled estimate of 13.8% (95% CI: 7.7%–23.6) in the south and 2.5% (95% CI: 0.1%–48.1%) in the north.

**Acute and chronic hepatitis E (RNA).**   As before, we divided the RNA group into those that use a total sample and in those that used only positive samples.

*The total sample for RNA analysis*. In 23 studies that presented RNA results from the total sample (24 results reported altogether), we found ranges of seroprevalence from 0%–54.5% with median of 0. All but one study reported RNA-seroprevalence under 10.4%. The one outstanding study [36] reported a seroprevalence of 54.5% among pediatric patients in Mexico; in contrast, IgG (IgM) seroprevalence in this group was 3% (6%).

From the other studies, nonzero seroprevalences were reported for cirrhosis patients in Argentina [45] (21%), renal transplant recipients in Brazil (10.4% [46] and 3.1% [47]), and patients with acute viral hepatitis in Argentina [48] (6.3%) and Cuba [49] (8.5%). For more details, see Fig 20, page 78 in S1 File.

*RNA among positive samples*. In 23 studies that reported information related to RNA positive results among IgG and/or IgM positive samples (26 results reported altogether), we found ranges of seroprevalence from 0%–100%, with median of 0%.

In eight cases, the underlying sample size was less than 10 and in two cases as low as one sample. Leaving these cases out, the proportion of RNA positive results ranged from 0%– 60% (median of 0%). Non-zero proportions were observed especially for people with acute viral hepatitis (Fig 21, page 79 in S1 File).

## Risk factors and route of transmission

Of the 142 included studies, 96 mentioned risk factors related to the HEV seropositivity (Table 5, page 37 in S1 File). Of those, 41 studies reported a measure of association such as Odds Ratios (OR, n = 39) or Prevalence Ratios (PR, n = 2).

**Increasing age.**   In the general population, increasing age is associated with HEV infection (OR 1.06 (1.05–1.06)) [50], increasing from 30 years [51], increasing age (OR 3.50 (1.39–8.87)) with high prevalence in older than 46 years old group [52], and among those aged between 41–60 years (OR 3.2 (1.09–9.7)) [53]. In children and young adults, seroprevalence was higher in people between 26 to 29 years of age compared to children below 5 years (OR 15.50 (2.17– 110.90)) [54].

Higher age is also a risk factor among blood donors, with a 4-fold increase odds of having HEV at an advanced age (45–59 years) than at younger age (OR 3.96 (1.54–10.22)) [55], and with a 3-fold increase odds for age $\geq$ 50 against <50 (OR 3.33 (1.11–9.95)) [56].

In the exposed population, being older than 30 (OR 3.61 (1.31–9.94)) [57] or older than 45 years old (OR 3.2 (1.1–10.7)) [58] was associated with HEV.

In the occupational group, US service members older than 35 years had higher odds to be HEV positive (OR 2.9 (0.9–8.8)) [59], and waste pickers older than 40 years were more likely to be HEV positive (OR 5.2 (1.5–17.5)) [60]. HEV seropositivity was associated with increasing age in waste pickers by OR 6.52 (1.95–21.78) [61] and in Mennonites by OR 1.05 (1.00–1.09) [62] in comparison with the control group.

Higher age also increases odds for HEV positivity in people with acute hepatitis (OR 2.04 (1.09–3.79) for those ≥ 60 years against <60 [63], and OR 1.5 (1.03–1.07) per-year increase) [64], and in those with immune deficiency (adjusted OR, 3.34 (1.54–7.24) for age ≥ 60 against <60 [65], and adjusted OR 1.05 (1.02–1.09) per 1 year increase) [66].

In rural population, increasing age, for each 1-year increase, was associated with HEV sero-positivity (OR 1.03 (1.01–1.05) [67], OR 1.04 (1.04–1.05) [68], OR 1.05 (1.04–1.07) [69]), especially in the 21–30 [67], and in the 41–50 age groups [70].

**Sex.**   In the general population, it was found that females had a higher prevalence than males (OR 1.2 (1.06–1.38)) [50], and those young females (6–39 years) were found to have higher HEV seropositivity compared to males (OR 1.44 (1.03–2.02)) [71]. In contrast, one study found that males had a higher anti-HEV IgG prevalence rate than females [72].

In the immunosuppressed group, being male was a risk factor for HEV (adjusted OR 2.07 (1.17–3.66)) [66], as well as for the viral hepatitis group (OR 2.10 (1.39–3.19)) [64].

**Ethnicity.**   For the general population, three studies referred to this factor. Two studies in USA found that non-Hispanic-Asian ethnicity was associated with HEV seropositivity when comparing with other groups (OR 1.69 (1.12–2.56)) [50]. In addition, non-Hispanic black people had lower rates of HEV infection than white participants (OR 0.60 (95% CI 0.46–0.79)) [71]. Moreover, place of birth was associated as an HEV factor [73, 74]. One study found that people born outside the United States were more likely to be HEV positive (OR 1.78 (1.23–2.58)) [71].

In Alaska, USA, native Americans were less likely to be seropositive to HEV than non-native people [75]. Moreover, Talarmin, A et al. also documented that seroprevalence rates differed significantly between ethnic groups, with higher adjusted-per-age rates among Brazilians (OR 4.8 (1.9–11.9), Chinese and Hmongs (OR 4.0 (1.5–10.9), and Haitians (OR 3.3 (1.1–9.9) when comparing against Creole [76].

Among viral hepatitis patients, being born outside the United States or Canada (OR 2.09 (1.03–4.25)), or being Asian and less educated people living in Canada or in USA (OR 2.92 (1.42–6.01)) were also documented as a risk factor for HEV seropositivity [64].

**Immunosuppressed system.**   Comorbidities such as cirrhosis (OR 8.95 (4.0–15.99)) [45], other liver diseases (adjusted OR 7.78 (3.43–17.64)) [66], or having low CD4 counts (OR 0.96 (0.94–0.99) for a CD4 increase of 10 counts) [77] increased the odds of having HEV. Moreover, one study demonstrated an association between the risk of post-transplant HEV infection and graft rejection (OR, 14.2 (1.26–160.0)) [78].

**Occupation.**   People occupationally exposed to pigs have more risk to be HEV positive than those without exposure (PR 2.42 (1.66–3.53)) [79]. Moreover, swine veterinarians were 1.51 times more likely to be anti-HEV positive than normal blood donors when tested with swine HEV antigen (CI 95% 1.03–2.20) [80]. Having HEV antibodies is associated with increasing years of occupation in a slaughterhouse (OR 8.81 (1.28–60.34), for employment = 20 years) [81]. Especially raising pigs, within waste pickers, was associated with being HEV positive (OR 12.01 (1.48–97.26)) [61].

**Food consumption practices.**   Consumption of meat or self-grown products is a potential risk factor for HEV in the general population. People that ingested self-grown food had higher

odds to have positive HEV than those who did not (OR 1.87 (1.41–2.48)) [71]. Consuming liver or other organ meats more than once per month (OR, 1.38 (1.01–1.88)) [82], or consuming meat more than 10 times per month increased the odds of being HEV positive [59]. Consumption of undercooked meat is related to the increased odds of being HEV positive in undergraduate and veterinary students (OR 12.9 (1.71–97.19)) [83].

In the immunocompromised group, fish consumption was a risk factor (OR 9.33 (2.07–42.2)) [84]. Moreover, a positive association between alcohol consumption and HEV seropositivity in immunocompromised patients (OR 3.43 (1.39–8.49)) [45] was reported.

**Rurality and low education.**   Other risk factors for infection in the general population were living in rural communities (OR 1.63 (1.16–2.30)) [54], and having a low educational level (OR 2.08 (1.04–4.17) [54]; PR 1.76 (1.02–3.04) [85]). Dwelling in a rural settlement for more than 5 years was associated with HEV seropositivity (OR 3.4 (1.2–9.6)) [86].

For patients with viral hepatitis additional contact with pigs (OR 1.99) [63], having low education level, living in a rural area, or having poor sanitation practices were potential risks for HEV infection [41, 87, 88].

**Sanitation conditions.**   In the general population, five studies found water and sanitation as a potential factors associated with HEV infection. Having a source of tap water lower the odds of having HEV (OR 0.78 (95% CI, 0.63–0.97)), while having any pet in the household (OR 1.19 (1.01–1.40), including dogs and cats), or having a dog in the household (OR 1.22 (1.04–1.43)) increases the odds [82]. Another study found higher HEV prevalence in people that have their water supply outside rather than inside their home (OR 3.15 (1.38–7.17)) [89].

In the rural population, consumption of untreated water (OR 1.92 (1.06–3.46)) and availability of water at home (OR 1.87 (1.07–3.27)) were factors associated to HEV [68].

For the immunocompromised group, piped water availability (OR 0.08 (0.01–0.66)) [90] decreases the odds of being HEV seropositive.

**Inner-country residence and travelling.**   In the USA, blood donors from the Midwest were more likely to be antibody positive compared to the rest of the country (OR 2.23, (1.92–2.88)) [91]. In Brazil, the highest prevalence was observed in the Central (OR 6.93, (1.15–41.61)) and South-Central (6.28 (1.22–32.22)) among the demographic zones in the city of Sao Paulo (with Northeast as the reference) [55].

Fearon, MA et al. reported that Canadian blood donors with a history of living outside Canada (OR 2.9, (1.56–5.32)) or contact with farm animals (OR 1.5 (1.01–2.28)) were associated with HEV seropositivity [92]. Moreover, seropositive blood donors were more likely to have travelled to endemic regions than seronegative ones (OR 3.4, (1.0–14.5)) [93]. Another study also found that national trips within Mexico were associated with HEV exposure (OR 5.38 (1.02–28.16)) [94].

**Pregnancy.**   Three studies have reported that HEV was associated with pregnancy (OR 3.5 (1.1–10.5)) [95], adjusting also for age and place of residence, or with having more than three pregnancies (OR 1.69 (1.04-2.75)) [96]. Moreover, Tissera, G et al. found an association between low age of a pregnancy (≤25 years old) and HEV seroprevalence (OR 3.2 (1.2–8.9)) [95].

**Transmission route.**   The main route of transmission mentioned in the studies was fecal-oral; [69, 97] one study found blood transfusion history as a risk factor [68].

## Genotyping

Twenty studies reported genotypes (Table 6, page 59 in S1 File). The most frequently studied genotypes in the Americas were 1 (Argentina, Venezuela, USA, Mexico, and Cuba) and 3

## HEV Genotype in the Americas

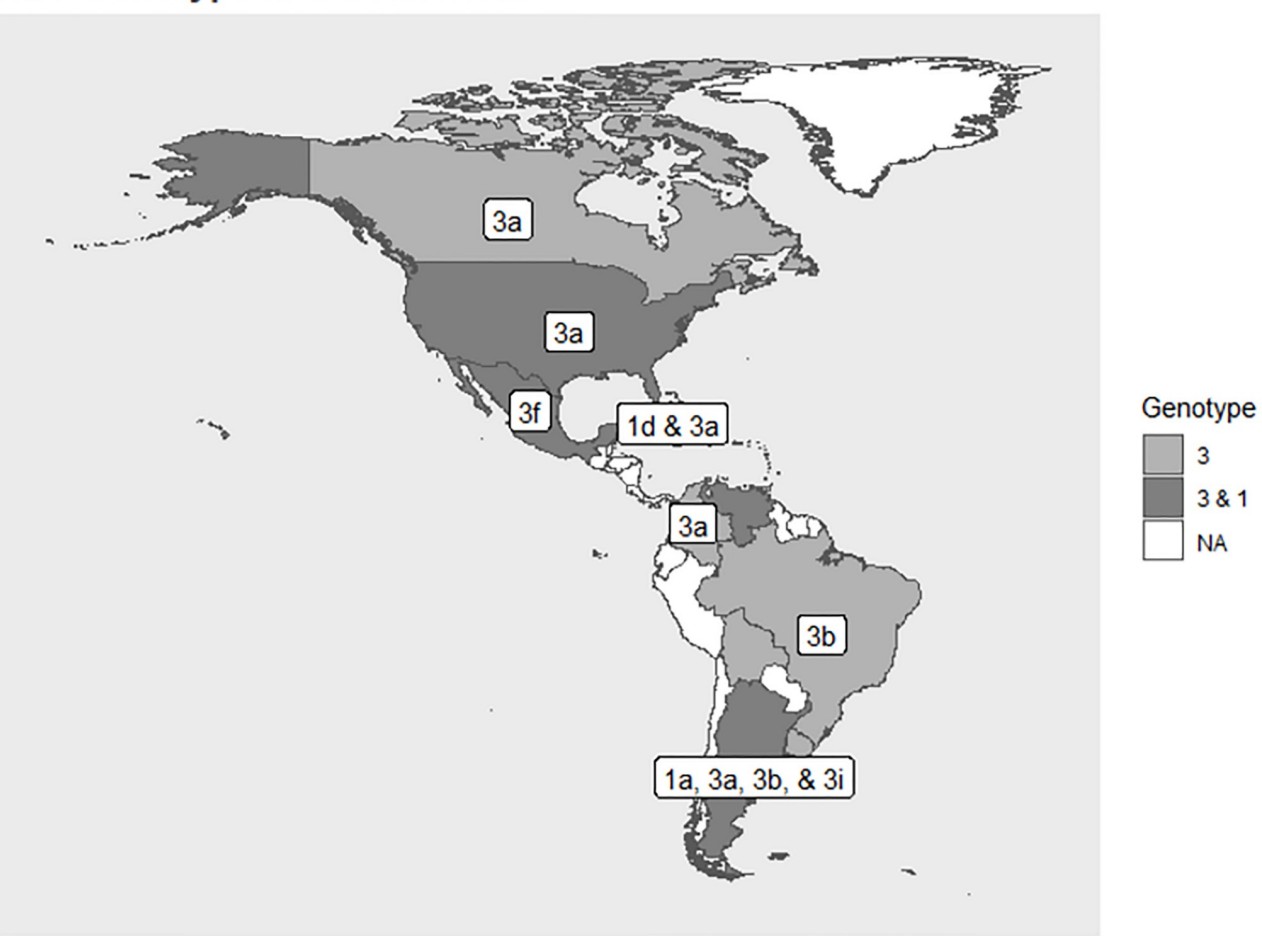

**Fig 3. HEV genotype distribution in the Americas according to the included studies.**

(Argentina, Brazil, Canada, Colombia, Venezuela, Bolivia, Mexico, USA, Cuba, and Uruguay) (Fig 3). Genotype 3 is more prevalent in this region and specifically the subtype 3a.

## Discussion

Our systematic review and meta-analysis summarized the seroprevalence of the hepatitis E virus and its variation according to different serological markers, population groups, and regions. Anti-HEV IgG, which indicates past HEV infection, was the most studied antibody with an overall seroprevalence of 7.7% (95% CI 6.4%–9.2%). Nevertheless, this estimate is influenced by the composition of studied populations as reported in the literature, since the seroprevalence seems to differ in certain subpopulations. For example, our analysis found a higher anti-HEV IgG seroprevalence in people exposed to pigs (13.8% (7.6%–23.6%)), exposed population (11.4% (5.4%–22.6)), and rural population (11.5% (5.4%–22.9%)). Risk factors for HEV prevalence that we identified from published data were age (highest in people over 30 years), contact with pigs and meat products, and some socioeconomic factors such as low sanitation access and low education level. Finally, we found predominance of genotype 1 and 3 in the American continent.

We found high variability in the seroprevalence estimates within regions of the Americas, countries, and within the considered population subgroups. A detailed inspection of the IgG seroprevalences reported from Brazil (Fig 13, page 72 in S1 File) suggests that this heterogeneity may be partly due to the different assays used. For example, the seroprevalences obtained using a test from Wantai tend to be higher than those obtained by a test from Abbott Laboratories. Pooling only results obtained by one test would yield in Brazil a seroprevalence of 15% (95% CI: 11%– 21%) in the former and of 4% (95% CI: 2%– 7%) in the latter case. This difference seems to be in line with the recently found low specificity (75%) for the Wantai test [98]. However, in our analysis, we do not correct for test characteristics, since the seroprevalence studies often do not report the numbers relevant for their investigations. Nevertheless, accounting for the test performance would be necessary to obtain more reliable seroprevalence estimates. Other studies also identified the employed assays as an important source of variability, together with the variance between regional areas, and the study design [99].

Our pooled seroprevalence estimate and the observed heterogeneitz may be compared to findings from other regions. Data from Europe indicates that the anti-HEV IgG seroprevalence rate ranges from 0.6% to 52.5% with high heterogeneity [99], with an overall anti-HEV IgG seroprevalence of 19% (95% CI 14%–26%) [16]. As in our review, the European HEV seroprevalence is higher in individuals exposed to swine/wild animals and increases with age [99]. In the Middle East and in the Eastern Mediterranean region, the anti-HEV antibody IgG seroprevalence was 12.17% (95% CI 11.79%–12.57%), and 11.81% (95% CI 11.43%–12.21%), respectively [100]. In terms of general population, global estimates for anti-HEV IgG found a seroprevalence of 12.47% (95% CI 10.42% - 14.67%; $I^2$ = 100%) [13].

Considering differences between countries and regions of the Americas, the pooled seroprevalences were comparable, with perhaps the exception of Bolivia, Peru and Colombia showing a higher anti-HEV IgG seroprevalence (Fig 2). However, the overall number of studies available for these three countries was rather low and consequently not all subpopulations were represented in the results. Thus, the apparent higher seroprevalence does not have to indicate a higher background seroprevalence in these countries. A similar limitation applies to the formally observed difference between North and South America within the immunodeficiency subgroup. The difference may reflect the particular choices for the studied populations within each region, since the immunodeficiency subgroup has been defined rather broadly. Other studies have found differences in the estimated human HEV seroprevalence in industrialized countries [16] and between European countries [99].

In terms of the risk factors reported by the studies, we identified increasing age as being associated with higher prevalence through all the population groups, especially in those over 30 years. This suggests a lifelong cumulative exposure to the virus [15, 101, 102]. Ditah et al. had suggested that there is a higher probability of exposure to HEV in those people that live longer, and for that reason the prevalence increased with age [73]. In terms of aging, there are multiple factors that affect disease incidence and mortality such as young age at exposure, the accumulation over time of exposure, and the effect of aging on immune function, genomic instability, and other aging processes [103]. In chronic diseases mainly, the exposition to a certain risk factor occurs through adult life, childhood or prenatally, and its biological effects will be cumulative and possibly advanced at older ages [104].

Higher anti-HEV seropositivity is related to the occupational population with contact with pig or with pig products [13, 102, 105]. This is relevant not only in low- and middle-income countries, but also in high-income countries because it confirms its relationship with zoonotic transmission and contaminated food. It can be prevented with the implementation of medical and veterinary health strategies, surveillance and disease control activities, and safe food handling practices and procedures. People can also avoid the consumption of raw or undercooked

products. Likewise, it will be important to perform studies in populations exposed to pigs and by assessing other potential risks and their contribution to HEV seroprevalence in those slaughterhouses and meat companies.

Based on our literature review, we conclude that more than ethnicity and country of birth or place of residence, the main risk factors are related to the socio-economic, education, and sanitary conditions that some populations have, especially those rural and indigenous communities where the fecal-oral transmission is more likely. Those areas that have compromised economic conditions have higher odds of exposure to animals and to contaminated surfaces such as soil or water [13]. Other authors have reported that there is evidence that premature deaths, having communicable diseases, or being affected by some pathogens is attributable to socioeconomic inequality [106–108]. In China for example, a higher prevalence of HEV in ethnic groups has been reported, especially in areas with mixed farming of domestic animals, or overpopulated and less developed areas [109, 110].

We documented that both genotype 1 and 3 are present in the Americas, which is consistent with the findings that fecal-oral transmission among humans as well as infections of zoonotic origin play a role in the Americas. This also indicates that transmission routes in the Americas are apparently very similar to high-income countries of other continents where the genotype 3 predominates [9, 111, 112]. This supports the evidence that HEV strains circulating among swine and humans are closely genetically related, supporting as well the transmission of HEV through consumption of raw or undercooked meat products [4]. Genotype 3 is higher documented in the region and specifically the subtype 3a. One European study has mentioned that subtype 3a has a worldwide presence and can be found in samples from humans, pigs, and wild animals, and especially in American samples it could be an indigenous subtype [113]. Regarding other subtypes, 1a is more prevalent in Asia, and 1d is found in Africa [4]. Subtypes 3a and 3b circulate in the United States and Japan, and 3f and 3i circulate in Europe [4]. Although there are different HEV-3 subtypes, there is no clear evidence of the clinical significance of infection based on the subtype. For example, European studies have found that the risk of HEV-3-infected patients being hospitalized varied with the subtype and that patients infected with subtype 3c were at lower risk of hospitalization than those infected with subtypes 3f or 3e [114, 115].

Besides the seroprevalence reported from the general population, specific patient groups were studied for acute and recent HEV infections. The IgM seroprevalence was higher (Q-test, p = 0.006) in patients with confirmed or suspected viral hepatitis (pooled seroprevalence 5.5%, 95% CI: 2.0%– 14.3%) when compared to the other subgroups (pooled seroprevalence 0.9%, 95% CI: 0.6%– 1.5%). Similarly, in two studies an RNA positivity of 6.3% [48] and of 8.5% [49] was reported in acute viral hepatitis patients. This suggests that in suspected cases of acute hepatitis it would be important to look also for hepatitis E virus, as this agent can be the cause of the disease.

To our knowledge, this is the first systematic review that focused on the Americas and the seroprevalence of HEV. The strengths are the broad search of data and the inclusion of studies without restrictions, for example by language, as well as the critical and systematic assessment of available data. Our focus was on HEV in different subgroups and the analysis according to different serological markers such as total antibodies, IgG, IgM, or RNA. We also performed a robust statistical analysis considering multiple options in terms of proportions by region, population groups, and countries.

Aiming at including all relevant evidence available, the main limitation of our approach results from the heterogeneity of studies in terms of methods and effect size. As shown in the heterogeneity and in the publication bias analysis (Figs 1 and 2, pages 60–61 in S1 File), we found that there are important differences between the studies not only in the sample size but

also in the methods and analysis, and in the small study effect. We studied the reasons behind the high heterogeneity between the studies. Nevertheless, we could not explain it, but we suspect that it could be because: 1) difference between the assays and their specificity and sensitivity, 2) difference between the years of the study, 3) difference between countries and within the regions and cities from the countries. Therefore, due to this heterogeneity, we consider our approach more explorative, and we are aware that our findings are to some extent influenced by bias and error.

Our results give us some ideas to develop. We can highlight gaps in evidence in the area of acute hepatitis and pig related exposure. It is important to further investigate the causality of HEV in acute hepatitis cases and outbreaks and in those people that work with pigs or meat products, and to assess attributable fractions of HEV. We recommend including testing for HEV as a standardized practice in suspected hepatitis cases and outbreaks when financially possible. We also consider measuring the real risk of infection that people who work with pigs or meat products have.

In conclusion, the seroprevalence of HEV varies between certain population groups, serological markers, and within regions and countries of the Americas. The main risk factors involved in the infection are increasing age, contact with pig or pig products, and poor socioeconomic conditions. This is in line with the findings that genotype 1 and 3 are the two most prevalent genotypes in the Americas and further supports the transmission of HEV as zoonotic and as human fecal-oral origin. Our results underline that water sanitation, occupational health, and food safety must be priority components for the prevention of HEV infections.

## Supporting information

**S1 File. Supplementary material.**
(PDF)

**S2 File. PRISMA checklist.**
(PDF)

## Author Contributions

**Conceptualization:** Nathalie Verónica Fernández Villalobos, Barbora Kessel, Jördis Jennifer Ott, Berit Lange.

**Data curation:** Nathalie Verónica Fernández Villalobos, Barbora Kessel, Isti Rodiah.

**Formal analysis:** Nathalie Verónica Fernández Villalobos, Barbora Kessel.

**Investigation:** Nathalie Verónica Fernández Villalobos.

**Methodology:** Nathalie Verónica Fernández Villalobos, Barbora Kessel.

**Supervision:** Jördis Jennifer Ott, Berit Lange, Gérard Krause.

**Validation:** Barbora Kessel, Isti Rodiah.

**Visualization:** Nathalie Verónica Fernández Villalobos, Barbora Kessel.

**Writing – original draft:** Nathalie Verónica Fernández Villalobos.

**Writing – review & editing:** Nathalie Verónica Fernández Villalobos, Barbora Kessel, Isti Rodiah, Jördis Jennifer Ott, Berit Lange, Gérard Krause.

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
