## [Decision Letter · Decision Letter 0]

4 Apr 2022

PONE-D-22-06563Seroprevalence of Hepatitis E Virus infection in the Americas: estimates from a systematic review and meta-analysisPLOS ONE

Dear Dr. Fernández Villalobos,

Thank you for submitting your manuscript to PLOS ONE. After careful consideration, we feel that it has merit but does not fully meet PLOS ONE’s publication criteria as it currently stands. Therefore, we invite you to submit a revised version of the manuscript that addresses the different points raised during the review process.

We look forward to receiving your revised manuscript.

Kind regards,

Isabelle Chemin, PhD

Academic Editor

PLOS ONE

Journal Requirements:

"The research was funded by intramural funds of the HZI, and the main author has a scholarship by Studienstiftung des deutschen Volkes. All authors confirm full access to all the data in the study and accept responsibility to submit for publication."

Reviewers' comments:

Reviewer's Responses to Questions

**Comments to the Author**

1. Is the manuscript technically sound, and do the data support the conclusions?

Reviewer #1: Yes

Reviewer #2: Yes

Reviewer #3: Yes

2. Has the statistical analysis been performed appropriately and rigorously? 

Reviewer #1: Yes

Reviewer #2: No

Reviewer #3: Yes

3. Have the authors made all data underlying the findings in their manuscript fully available?

Reviewer #1: Yes

Reviewer #2: Yes

Reviewer #3: Yes

4. Is the manuscript presented in an intelligible fashion and written in standard English?

Reviewer #1: Yes

Reviewer #2: Yes

Reviewer #3: Yes

5. Review Comments to the Author

Reviewer #1: Fernandez Villalobos et al. conducted a great and very interesting study on the seroprevalence, risk factors, transmission routes and genotypes of HEV in America.

This manuscript is well written but there are a few points to review to improve the quality of the work. I therefore recommend it for publication after considering the following comments.

Introduction

L85-93: The review authors provide a wide summary of previous systematic reviews on the topic which is supported by Supplementary Table 2. With recent reviews having up to 68 studies (Horvaits, T et al 2018), 45 studies (Li, P et al 2020), and 42 studies (Wilhelm, B et al 2019) from America. However, the authors of this review were able to include up to 142 studies. This suggests different criteria for this review compared to previous ones. I suggest that the authors further strengthen rationale of this review by highlighting the specificities of their study on criteria such as the population, the diagnostic method, the research strategy, etc.

Methodology

I suggest authors to mention in the main manuscript the PRISMA checklist which was provided as a supplementary file.

I suggest authors to hand-search for additional studies in the bibliography of previously published related reviews?

I suggest that the authors specify the exclusion criteria. An outbreak investigation for example may be a case report, so it is good to clarify that case reports are excluded.

I suggest adding a paragraph on reference management and study selection.

L136: I suggest indicating how the two investigators who assessed the risk of bias proceeded to deal with disagreements.

Results

I suggest the authors to add a legend to explain the black and red text coding colors in Table 4 of the Supplementary Material.

L272-274. I suggest indicating the figure that relates to these findings in the main manuscript.

Reviewer #2: Abstract

Background:

1. Please remove (Literature indicates sporadic evidence on HEV in the Americas).

2. Please remove (Our systematic review and meta-analysis).

Methods:

1. Please remove (Our systematic review and meta-analysis (registration number in PROSPERO CRD42020173934) included peer-reviewed and published data on the seroprevalence of HEV in humans in the Americas).

2. Please add the names of the databases and the duration of the search.

3. Please add the complete name of the GLMM.

Main text:

1. Some sentences need references.

2. The background is long, please short it.

Method:

1. A bout publication bias, I find not a plan about it. I think, the authors do not have enough information about the publication bias.

2. The method section is weak. Some sentences repeated about meta-analysis.

Discussion

1. The discussion of the article needs serious revisions. Findings should be emphasized in the discussion.

Reviewer #3: This manuscript has well illustrated a scenario of hepatitis E virus infection in the Americas. I wonder whether the authors have identified and excluded the publications which might share some identical data, especially in different languages.

6. PLOS authors have the option to publish the peer review history of their article (what does this mean?). If published, this will include your full peer review and any attached files.

Reviewer #1: No

Reviewer #2: **Yes: **Masoud Behzadifar, Lorestan University Medical of Sciences.

Reviewer #3: **Yes: **Yihan Lu

---

## [Author Response · Author response to Decision Letter 0]

28 Apr 2022

Reviewer #1 

Fernandez Villalobos et al. conducted a great and very interesting study on the seroprevalence, risk factors, transmission routes and genotypes of HEV in America.

This manuscript is well written but there are a few points to review to improve the quality of the work. I therefore recommend it for publication after considering the following comments.

#1 - Introduction

L85-93: The review authors provide a wide summary of previous systematic reviews on the topic which is supported by Supplementary Table 2. With recent reviews having up to 68 studies (Horvaits, T et al 2018), 45 studies (Li, P et al 2020), and 42 studies (Wilhelm, B et al 2019) from America. However, the authors of this review were able to include up to 142 studies. This suggests different criteria for this review compared to previous ones. I suggest that the authors further strengthen rationale of this review by highlighting the specificities of their study on criteria such as the population, the diagnostic method, the research strategy, etc.

Response: We really appreciate this comment. We agreed with you because we were able to include further evidence in our study thanks to the search strategy and no language restriction. We have specified this in the introduction line 87.

#2 - Methodology

I suggest authors to mention in the main manuscript the PRISMA checklist which was provided as a supplementary file.

Response: We have included this point in line 101.

#3 - I suggest authors to hand-search for additional studies in the bibliography of previously published related reviews?

Response: Thank you for this valuable comment. Indeed, during the search we applied the snowball method in the existing reviews to verify that we have included all the studies that were available. After corroborating this, we did not include any extra study, so we did not mention this in the manuscript. This is now corrected in line 104.

#4 - I suggest that the authors specify the exclusion criteria. An outbreak investigation for example may be a case report, so it is good to clarify that case reports are excluded.

Response: I apologize for this misunderstanding. In the qualitative review, we included all the evidence available, including outbreak investigation reports. For the quantitative analysis (meta-analysis), we excluded studies with low sample size (line 194) and one report with 100% of positive cases (line 362). 

#5 - I suggest adding a paragraph on reference management and study selection.

Response: Thank you for the suggestion. We added this information now: line 106 and line 121 respectively.

#6 - L136: I suggest indicating how the two investigators who assessed the risk of bias proceeded to deal with disagreements.

Response: Thank you for this comment. We had a third reviewer available in case of disagreements; however, we solved the conflicts between the two investigators. Therefore, the third reviewer was not needed at the end. We have now specified this in line 142.

Results

#7 - I suggest the authors to add a legend to explain the black and red text coding colours in Table 4 of the Supplementary Material.

Response: We apologize for this missing information and have included the legend on the respective table (S1 Table 4)

#8 - L272-274. I suggest indicating the figure that relates to these findings in the main manuscript.

Response: Thank you for your comment. However, we prefer not to focus on the presentation of overall pooled estimates since these are influenced by the composition of the studies, and as we have seen differing levels of seroprevalences in different subpopulations. For that reason, we focused on presenting the pooled estimates from the subgroup analysis.

Reviewer #2

#1 - Abstract

Background:

1. Please remove (Literature indicates sporadic evidence on HEV in the Americas).

2. Please remove (Our systematic review and meta-analysis).

Response: Thank you for these observations. We agreed with these two points and deleted the content in line 25.

#2 - Methods:

1. Please remove (Our systematic review and meta-analysis (registration number in PROSPERO CRD42020173934) included peer-reviewed and published data on the seroprevalence of HEV in humans in the Americas).

2. Please add the names of the databases and the duration of the search.

3. Please add the complete name of the GLMM.

Response: Thank you for your comments. We have deleted the content in line 30, and included the names of the databases and date (line 29) and the name of the model (line 33).

#3 - Main text:

1. Some sentences need references.

2. The background is long, please short it.

Response: Thank you for these comments. We have strived to improve the quality of the text, accounting also for the comments from the other two reviewers (who requested adding some information). We also carefully checked that all statements in the text are appropriately referenced.

Method:

#4 - 1. A bout publication bias, I find not a plan about it. I think, the authors do not have enough information about the publication bias.

Response: Thank you for your comment. We have improved the respective paragraph in the manuscript to make our point clearer in line 181. 

We have the understanding that the publication bias usually refers to a bias resulting from the fact that some studies are not published due to the direction and magnitude of their findings (i.e. because their results are not statistically significant). In our opinion, for seroprevalence studies, the statistical significance does not play a central role. We believe that the chance of not publishing the results because of a particular value of the seroprevalence found in the data is negligible. However, given the differences in seroprevalence in different subpopulations, our analysis is definitely affected by the choice of the populations to be studied. This can be influenced by convenience considerations, by funding incentives, by existing cohorts etc. We presented the composition of the populations as found in the publications in S1 Fig 2, and we comment on how this affects our results in the Discussion part of our paper (line 670).

#5 - 2. The method section is weak. Some sentences repeated about meta-analysis.

Response: We appreciate your comment and have now revised the text, tried to improve the flow, and avoid any repetitions without impairing the clarity and completeness of the description of our analyses. We hope to have amended the places you had found problematic.

#6 - Discussion

1. The discussion of the article needs serious revisions. Findings should be emphasized in the discussion.

Response: Thank you for your outlook. We critically revised the Discussion section, reordered, and reformulated some parts of it. We hope that the current text makes clearer our findings, their implications, their relationship with previous findings and limitations.

Reviewer #3

This manuscript has well illustrated a scenario of hepatitis E virus infection in the Americas. I wonder whether the authors have identified and excluded the publications which might share some identical data, especially in different languages.

Response: Thank you for this relevant comment. As we included articles in Spanish and English in the majority of cases, people can think about this problem. However, in this respect we did not find any exact translations or identical data between studies. Nevertheless, within one language we found several studies sharing the same samples as detailed in S1 Table 4. Here we always included only one representative in the quantitative analysis; the exclusions are specified in S1 Table 4 in red.

---

## [Editor Report · Decision Letter 1]

18 May 2022

Seroprevalence of Hepatitis E Virus infection in the Americas: estimates from a systematic review and meta-analysis

PONE-D-22-06563R1

Dear Dr.  Fernández Villalobos,

We’re pleased to inform you that your manuscript has been judged scientifically suitable for publication and will be formally accepted for publication once it meets all outstanding technical requirements.

Kind regards,

Isabelle Chemin, PhD

Academic Editor

PLOS ONE
---

## [Editor Report · Acceptance letter]

20 May 2022

PONE-D-22-06563R1 

Seroprevalence of Hepatitis E Virus infection in the Americas: estimates from a systematic review and meta-analysis 

Dear Dr. Fernández Villalobos:

I'm pleased to inform you that your manuscript has been deemed suitable for publication in PLOS ONE. Congratulations! Your manuscript is now with our production department. 

Kind regards, 

on behalf of

Mrs Isabelle Chemin 

Academic Editor

PLOS ONE